# TASK CALIBRATION: CALIBRATING LARGE LANGUAGE MODELS ON INFERENCE TASKS

## ABSTRACT

Large language models (LLMs) have exhibited impressive zero-shot performance on inference tasks. However, LLMs may suffer from spurious correlations between input texts and output labels, which limits LLMs' ability to reason based purely on general language understanding. In other words, LLMs may make predictions primarily based on premise or hypothesis, rather than both components. To address this problem that may lead to unexpected performance degradation, we propose *task calibration* (TC), a zero-shot and inference-only calibration method inspired by mutual information which recovers LLM performance through task reformulation. TC encourages LLMs to reason based on both premise and hypothesis, while mitigating the models' over-reliance on individual premise or hypothesis for inference. Experimental results show that TC achieves a substantial improvement on 13 inference tasks in the zero-shot setup. We further validate the effectiveness of TC in few-shot setups and various natural language understanding tasks. Further analysis indicates that TC is also robust to prompt templates and has the potential to be integrated with other calibration methods.

## 1 INTRODUCTION

Large language models (LLMs) (Touvron et al., 2023; Chowdhery et al., 2024; Abdin et al., 2024) have demonstrated strong generalization ability to excel in a wide range of downstream tasks. In particular, prompt-based learning has been an effective paradigm for LLMs, enabling zero-shot or few-shot learning (Brown et al., 2020; Liu et al., 2023). Ideally, an LLM with advanced language understanding capabilities could perform natural language inference (NLI) in a zero-shot setting without relying on annotated examples. However, research has shown that zero-shot capabilities of models on inference tasks are currently constrained by the presence of spurious correlations that often lead to biased prediction (McKenna et al., 2023).

To mitigate spurious correlations, previous work (Zhao et al., 2021; Holtzman et al., 2021; Fei et al., 2023; Han et al., 2023; Zhou et al., 2024) has explored model calibration, which reweighs output probabilities based on various bias estimators. However, existing calibration methods fall short of addressing the bias that stems from LLMs' reliance on either the premise or hypothesis for prediction (McKenna et al., 2023), which we call preference bias. This limits their capacity to generalize in inference tasks. Figure 1 shows an example from QNLI dataset (Rajpurkar et al., 2016), where the task is to determine whether a given context sentence contains the answer to a given question. We observe that the model prediction is incorrect because it relies excessively on the question itself when making the prediction in this example.

Motivated by this observation, we propose **task calibration** (TC), a zero-shot and inference-only calibration method. Our work is inspired by mutual information (Tishby et al., 1999; Peng et al., 2005), which measures how much one random variable tells us about another. Intuitively, for a specific task, proper use of mutual information can reveal how much more informative the combined presence of premise and hypothesis is concerning the label, compared to their individual presences. Based on this insight, we reformulate LLM inference by factoring out the probabilities of premise-only and hypothesis-only inputs. TC requires no annotated data and is easy to implement, involving only two extra inference stages using premise-only and hypothesis-only inputs for each sample. As shown in Figure 1, although the model's initial answer is incorrect, it finally makes

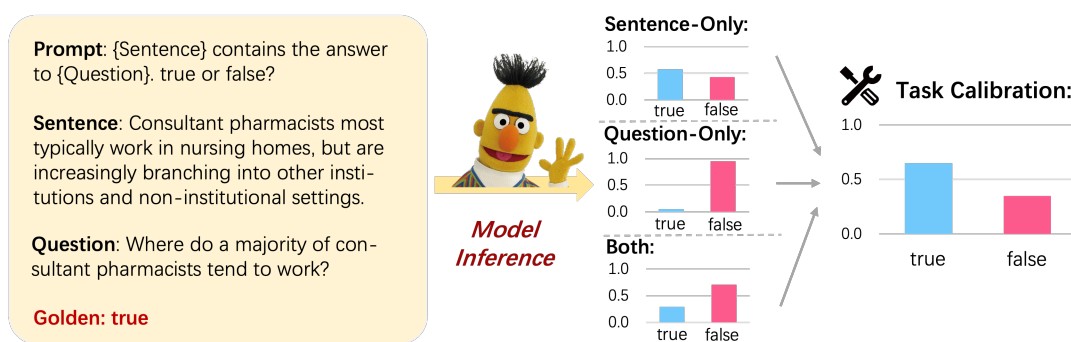

Figure 1: An example from QNLI dataset (Rajpurkar et al., 2016). *Sentence-Only*, *Question-Only* and *Both* indicate the inputs with only the sentence, question and using both components, respectively. While the initial model prediction is incorrect, potentially due to the influence of the hypothesis, we observe that task calibration finally leads to a correct prediction.

the correct prediction after task calibration, by using output probabilities derived from premise-only, hypothesis-only, and combined inputs.

Experimental results demonstrate superior performance of TC over other calibration methods in the zero-shot setup, showcasing a noteworthy boost of three different LLMs on 13 inference datasets. Specifically, TC outperforms the best-performing baseline in 12, 9 and 10 out of 13 datasets on the Mistral-7B-Instruct-v0.3, Llama-2-7B-chat and Phi-3-mini-4k-instruct models, respectively. In addition, TC is robust to various prompt templates, demonstrating its effectiveness in few-shot setups and 4 different natural language understanding (NLU) tasks such as sentiment analysis and hate speech detection. Finally, we find that the combination of TC and other calibration methods can yield better performance, which indicates their complementary strengths in fixing spurious correlations.

To summarize, our key contributions are as follows:

- We are the first to consider the synergistic effect of premise and hypothesis over their individual effects in model calibration.

- We propose task calibration (TC), a zero-shot and inference-only calibration method, which alleviates the bias in LLMs that arises from an over-reliance on either the premise or hypothesis for prediction.

- We show that TC achieves state-of-the-art performance on 13 inference datasets in the zero-shot setup. TC is robust to prompt templates, and also demonstrates its effectiveness in few-shot setups and 4 different NLU tasks.

## 2 RELATED WORK

**Spurious Correlations in Inference Tasks.** The issue of spurious correlations between labels and some input signals has attracted considerable attention in the NLP field. It has been shown that a model that only has access to the hypothesis can perform surprisingly well on NLI tasks, suggesting the existence of hypothesis-only bias within the datasets (Poliak et al., 2018; Gururangan et al., 2018; Tsuchiya, 2018; Glockner et al., 2018). Similar bias can be observed in QA (Kaushik & Lipton, 2018; Patel et al., 2021), fact verification (Schuster et al., 2019) and stance detection (Kaushal et al., 2021) tasks, where models can achieve remarkable performance without considering any question, evidence and target, respectively. Recently, McKenna et al. (2023) identify the attestation bias, where LLMs falsely label NLI samples as entailment when the hypothesis is attested in training data. In Section 4, we observe that, when provided with premise-only or hypothesis-only inputs, LLMs often struggle to predict *not_entailment*, and frequently make identical predictions with those using both components. This indicates the potential existence of preference bias that enables LLMs to perform inference without relying on both premise and hypothesis.

**Calibration of Language Models.** Previous attempts to mitigate spurious correlations include training a debiased model with residual fitting (He et al., 2019) or a debiased training set (Wu et al., 2022). However, these methods necessitate fine-tuning, and thus pose challenges for pursuing efficient LLMs. Zhao et al. (2021) propose contextual calibration (CC), which first estimates the bias of language models with a content-free test input, and then counteracts the bias by calibrating the output distribution. Holtzman et al. (2021) find that different surface forms compete for probability mass. Such competition can be greatly compensated by a scoring choice using domain conditional pointwise mutual information (DCPMI) that reweighs the model predictions. Fei et al. (2023) further identify the domain-label bias and propose a domain-context calibration method (DC) that estimates the label bias using random in-domain words from the task corpus. Han et al. (2023) propose prototypical calibration to learn a decision boundary with Gaussian mixture models for zero-shot and few-shot classification. Zhou et al. (2024) propose batch calibration (BC) to estimate the contextual bias for each class from a batch and obtain the calibrated probability by dividing the output probability over the contextual prior. In contrast, we tackle the problem from a different perspective of task reformulation, which mitigates bias while recovering model performance across challenging inference tasks.

## 3 EXPERIMENTAL SETUP

**Datasets.** We conduct experiments on 17 text classification datasets that cover a wide range of tasks. Specifically, for standard inference task, we consider natural language inference: RTE (Dagan et al., 2005), WNLI (Levesque et al., 2011), SciTail (Khot et al., 2018), CB (Marneffe et al., 2019), MNLI (Williams et al., 2018) and QNLI (Rajpurkar et al., 2016); stance detection: Perspectrum (Chen et al., 2019), IBM30K (Gretz et al., 2020), EZ-Stance (Zhao & Caragea, 2024), IAM (Cheng et al., 2022) and VAST (Allaway & McKeown, 2020); paraphrasing: PAWS (Zhang et al., 2019) and QQP. To indicate the effectiveness of TC on other tasks, we follow the experimental setting that adopts a textual entailment formulation in previous work (Yin et al., 2019; Ma et al., 2021) and additionally consider sentiment classification: SST-2 (Socher et al., 2013); offensive language identification: OffensEval (Barbieri et al., 2020); hate speech detection: HatEval (Barbieri et al., 2020) and HateSpeech18 (de Gibert et al., 2018). RTE, WNLI, CB, MNLI, QNLI and QQP datasets used for evaluation are drawn from the GLUE (Wang et al., 2018) and SuperGLUE (Wang et al., 2019) benchmarks. More details of these datasets can be found in Table 6 of Appendix. We use the test set for evaluation except for GLUE and SuperGLUE datasets, for which we use the full validation set for evaluation. Note that we exclude datasets such as OpenBookQA (Mihaylov et al., 2018) and NQ (Kwiatkowski et al., 2019), since we aim to assess LLMs' ability to reason based purely on general language understanding, not prior knowledge.

**Baselines.** We compare TC with the original LM and previous calibration methods, including CC (Zhao et al., 2021), DCPMI (Holtzman et al., 2021), DC (Fei et al., 2023) and BC (Zhou et al., 2024). These methods are discussed in Section 2 and their scoring functions are shown in Table 1. We follow the same setup with original papers in the implementation. For CC, we average the probabilities from three content-free inputs: 'N/A', '[MASK]', and the empty string. For DCPMI, we adopt the same domain premise (e.g., 'true or false? Answer:') on inference datasets. For DC, we sample the same number (i.e., 20) of random texts for estimating model's prior. For BC, we compute the correction log-probability once after all test samples are seen as suggested.

**Model and Implementation Details.** We conduct experiments mainly on three instruction-tuned models including Mistral-7B-Instruct-v0.3[1] (Jiang et al., 2023), Llama-2-7B-chat[2] (Touvron et al., 2023) and Phi-3-mini-4k-instruct (3.8B) [3] (Abdin et al., 2024). For all experiments, unless stated otherwise, we perform the evaluation in the zero-shot setting. In the few-shot setting, we use n = 1-4 example(s) sampled randomly from the training set to construct the context prompt and evaluate five times using different random seeds. The templates and label names used for all datasets can be found in Table 7 of Appendix. We conduct the evaluation on an NVIDIA RTX A6000 GPU for all models. Following prior work (Fei et al., 2023; Zhou et al., 2024), we use the accuracy as the evaluation metric except for stance detection datasets, for which we use the Macro-F1 score.

---

[1] https://huggingface.co/mistralai/Mistral-7B-Instruct-v0.3
[2] https://huggingface.co/meta-llama/Llama-2-13b-chat-hf
[3] https://huggingface.co/microsoft/Phi-3-mini-4k-instruct

## 4  PREFERENCE BIAS

Without loss of generality, we use NLI as the main target for discussion in this section and Section 5, despite that our method can be used in other tasks. NLI requires distinct types of reasoning (Condoravdi et al., 2003), with the ideal inference depending on both premise and hypothesis (Poliak et al., 2018). Here, we empirically demonstrate LLMs' *preference bias*, which refers to a model's tendency to perform inference tasks without relying on both the premise and the hypothesis. This bias may potentially lead to performance degradation on out-of-distribution inference tasks. McKenna et al. (2023) identify the *attestation bias*, which can be seen as a special case of preference bias where LLMs falsely associate the hypothesis with *entailment*.

We explore the preference bias from a novel viewpoint, i.e., we examine whether LLMs can accurately predict *not_entailment* when the premise or hypothesis is absent from the input. Specifically, we evaluate Mistral-7B-Instruct-v0.3 on binary NLI tasks RTE (Dagan et al., 2005), SciTail (Khot et al., 2018) and QNLI (Rajpurkar et al., 2016) datasets where outputs include *not_entailment* or *entailment*. Ideally, LLMs should be able to discern the absence of premise or hypothesis and make predictions on *not_entailment*. As shown in Figure 2, Mistral-7B-Instruct-v0.3 exhibits a tendency to associate premise-only or hypothesis-only inputs with labels other than *not_entailment*, as evidenced by the gap between the bars and the ideal value (i.e., 100%). It suggests the existence of spurious correlations (which we call preference bias) that can distract LLMs from relying on both premise and hypothesis when making predictions. In addition, the performance of LLMs on premise-only and hypothesis-only inputs varies across datasets. For example, Mistral-7B-Instruct-v0.3 exhibits superior performance in the premise-only setting for SciTail and performs better in the hypothesis-only setting for RTE.

Building upon the observation, we further investigate the correlation between incorrect LLM predictions (using both premise and hypothesis) and the labels derived from premise-only or hypothesis-only inputs. Results are shown in Figure 3. We observe that LLM predictions based solely on the premise or the hypothesis frequently align with incorrect predictions of using both components. For example, in the SciTail dataset, over 90% of incorrect LLM predictions align with the labels obtained from hypothesis-only inputs. It reveals that the LLM excessively relies on the premise or hypothesis alone when making predictions.

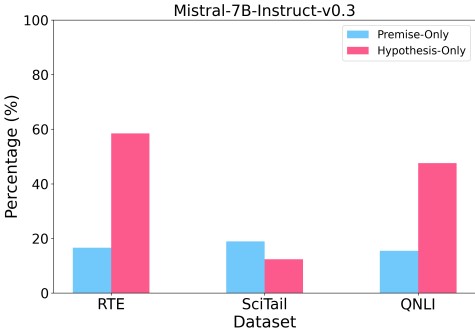

Figure 2: The percentage of LLM predictions on label *not_entailment* (NLI) with premise-only and hypothesis-only inputs. Higher value indicates low bias.

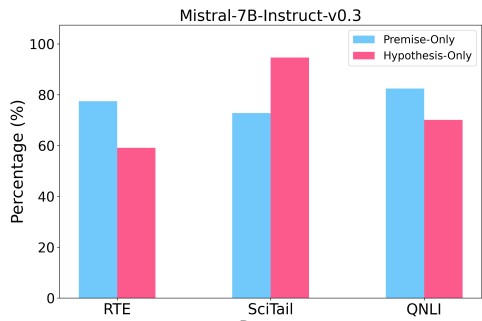

Figure 3: The percentage of erroneous LLM predictions that align with the labels derived from premise-only or hypothesis-only inputs. Higher value indicates high correlation.

## 5  TASK CALIBRATION

### 5.1  PROBLEM FORMULATION

Prompting has emerged as an effective strategy for LLMs to perform zero-shot inference with human instructions. For an NLI task, denoting a sentence pair $(x_p, x_h)$ and a possible label $y$ for inference tasks, LLMs make prediction by calculating: $\arg\max_{y \in \mathcal{Y}} p(y|x_p, x_h)$, where $\mathcal{Y}$ denotes the verbalizers that define the label set of C classes, and $p \in \mathbb{R}^C$ is the prediction probability.

Table 1: Comparison of scoring functions between task calibration (TC) and each calibration baseline on inference tasks. The example is selected from the RTE dataset (Dagan et al., 2005).

| *Text:* | *Baselines:* |
|---|---|
| **Premise** ($x_p$): Mount Olympus towers up from the center of the earth | **Probability** (LLM) $\arg\max_{y \in \mathcal{Y}} p(y\|x_p, x_h)$ |
| **Hypothesis** ($x_h$): Mount Olympus is in the center of the earth | **Contextual Calibration** (CC) $\arg\max_{y \in \mathcal{Y}} w p(y\|x_p, x_h) + b$ |
| **Template**: {} entails {}. true or false? Answer: | **Domain Conditional PMI** (DCPMI) |
| **Domain Text** ($x_{\text{domain}}$): true or false? Answer: | $\arg\max_{y \in \mathcal{Y}} \frac{p(y\|x_p, x_h)}{p(y\|x_{\text{domain}})}$ |
| **Random Text** ($x_{\text{rand}_1}$): {random in-domain text for the premise} | **Domain-context Calibration** (DC) $\arg\max_{y \in \mathcal{Y}} \frac{p(y\|x_p, x_h)}{p(y\|x_{\text{rand}_1}, x_{\text{rand}_2})}$ |
| **Random Text** ($x_{\text{rand}_2}$): {random in-domain text for the hypothesis} | **Batch Calibration** (BC) $\arg\max_{y \in \mathcal{Y}} \frac{p(y\|x_p, x_h)}{\frac{1}{N}\sum_{j=1}^{N} p(y\|x_p^j, x_h^j)}$ |
| | **Our Method: Task Calibration** (TC) $\arg\max_{y \in \mathcal{Y}} p(y\|x_p, x_h) \log\left(\frac{p(y\|x_p, x_h)^2}{p(y\|x_p)p(y\|x_h)}\right)$ |

## 5.2 MUTUAL INFORMATION IN CALIBRATION

To factor out the probability of specific surface forms, Holtzman et al. (2021) propose domain conditional PMI (DCPMI) to indicate the extent to which the input text is related to the answer within a domain. This concept is articulated in the context of inference tasks as follows:

$$\arg\max_{y \in \mathcal{Y}} \text{PMI}_{\text{DC}} = \arg\max_{y \in \mathcal{Y}} \log\left(\frac{p(y \mid x_p, x_h)}{p(y \mid x_{\text{domain}})}\right), \tag{1}$$

where $x_{\text{domain}}$ denotes a short domain-relevant string, which is fixed for a specific task. An example of $x_{\text{domain}}$ is shown in Table 1. Then, the mutual information of applying DCPMI to the task can be written as:

$$\text{MI}_{\text{DC}} = \sum_{x_p, x_h, y} p(x_p, x_h, y) \log\left(\frac{p(y \mid x_p, x_h)}{p(y \mid x_{\text{domain}})}\right). \tag{2}$$

However, DCPMI calibrates model predictions with content-free tokens (i.e., $x_{\text{domain}}$), which may introduce additional biases that lead to biased predictions (Zhou et al., 2024). Moreover, $\text{MI}_{\text{DC}}$ fails to take preference bias into considerations, which may account for the failures in Section 6.

## 5.3 REFORMULATION OF INFERENCE TASKS

Given two random variables $A$ and $B$, their mutual information is defined in terms of their probabilistic density functions $p(a)$, $p(b)$, and $p(a, b)$:

$$I(A; B) = \iint p(a, b) \log\left(\frac{p(a, b)}{p(a)p(b)}\right) da\, db. \tag{3}$$

$I(A; B)$ is a measure of the mutual dependence between $A$ and $B$, reflecting the reduction in uncertainty of one variable through knowledge of the other. Inspired by the concept of mutual information (Tishby et al., 1999; Peng et al., 2005), we introduce $I(X_p, X_h; Y)$ to indicate the joint dependency of inputs (i.e., premise and hypothesis) on the target class. Ideally, LLMs should depend on both premise and hypothesis to make predictions on inference tasks. However, as discussed in Section 4, LLMs with only $x_p$ or $x_h$ as input can still predict *entailment* on NLI datasets, indicating the existence of spurious correlations between labels and texts that may limit the reasoning ability of

LLMs. To mitigate the models' excessive reliance on solely $x_p$ or $x_h$ when making predictions, we propose task calibration (TC), which defines $\text{MI}_{\text{TC}}$ as follows:

$$\text{MI}_{\text{TC}} := I(X_p, X_h; Y) - \frac{1}{2}I(X_p; Y) - \frac{1}{2}I(X_h; Y)$$

$$= \sum_{x_p, x_h, y} p(x_p, x_h, y) \left[ \log \frac{p(y \mid x_p, x_h)}{p(y)} - \frac{1}{2} \log \frac{p(y \mid x_p)}{p(y)} - \frac{1}{2} \log \frac{p(y \mid x_h)}{p(y)} \right]$$

$$= \sum_{x_p, x_h, y} p(x_p, x_h, y) \log \left( \frac{p(y \mid x_p, x_h)}{\sqrt{p(y \mid x_p) p(y \mid x_h)}} \right), \tag{4}$$

where $p(y|x_p)$ and $p(y|x_h)$ denote the prediction probabilities of using only premise and hypothesis as input, respectively. Since Figure 2 reveals the presence of bias towards both premise-only and hypothesis-only inputs, we assign an equal weight of 0.5 to both components. $\text{MI}_{\text{TC}}$ quantifies the joint dependency of $X_p$ and $X_h$ on $Y$, beyond their individual dependencies. In essence, $\text{MI}_{\text{TC}}$ highlights the synergistic effect of $X_p$ and $X_h$ in predicting $Y$, rather than their separate contributions. Instead of directly using $\arg\max_{y \in \mathcal{Y}} p(y|x_p, x_h)$ as the scoring function, TC reformulates the inference tasks as:

$$\arg\max_{y \in \mathcal{Y}} p(y \mid x_p, x_h) \log\left( \frac{p(y \mid x_p, x_h)^2}{p(y \mid x_p) p(y \mid x_h)} \right). \tag{5}$$

Note that we remove the square root from Equation 4 for more natural expression. TC is an inference-only method that requires no fine-tuning and annotated data. It brings only two additional inferences of $p(y|x_p)$ and $p(y|x_h)$ for each sample. We compare the TC with previous calibration methods in Table 1. Unlike previous methods, which calibrate model predictions by either relying on content-free tokens or estimating contextual priors, TC mitigates the effects of spurious correlations by reducing LLMs' reliance on individual $x_p$ or $x_h$ through task formulation.

### 5.4 TASK CALIBRATION ON INFERENCE TASKS

As discussed in Section 3, our evaluation focuses primarily on NLI, stance detection and paraphrasing tasks. Concretely, $x_p$ and $x_h$ represent the premise and the hypothesis in NLI tasks, respectively. An example is shown in Figure 1, where Sentence and Question can be seen as the premise and the hypothesis, respectively. In stance detection tasks, $x_p$ and $x_h$ correspond to the text and the target (or claim), respectively. For example, the text "College exposes students to diverse people and ideas." can be considered as $x_p$ and the claim "College education is worth it." can be seen as $x_h$. Similarly, $x_p$ and $x_h$ represent different sentences in paraphrasing tasks. For instance, the queries "What was the deadliest battle in history?" and "What was the bloodiest battle in history?" can be seen as the $x_p$ and $x_h$, respectively.

## 6 EXPERIMENTS

### 6.1 MAIN RESULTS

**Zero-Shot Experiments on Inference Tasks.** We report the zero-shot performance of Mistral-7B-Instruct-v0.3, Llama-2-7B-chat and Phi-3-mini-4k-instruct across a diverse set of inference tasks in Table 2. Notably, TC consistently outperforms the original LLM (without calibration) across all datasets on all LLMs. In some cases, the absolute improvement can be over 40% and 20%, respectively, like Mistral-7B-Instruct-v0.3 on CB and Llama-2-7B-chat on SciTail in Table 2. It indicates that our proposed TC unleashes the potential of LLMs by mitigating spurious correlations that often lead to biased predictions. In addition, TC shows promising improvements over state-of-the-art calibration methods, surpassing them in 12, 9 and 10 out of 13 datasets on the Mistral-7B-Instruct-v0.3, Llama-2-7B-chat and Phi-3-mini-4k-instruct models, respectively. It is noteworthy that TC demonstrates stable performance improvements, in contrast to previous baselines which exhibit significant fluctuations in performance across tasks, often leading to frequent and notable performance degradation.

**Few-Shot Experiments.** While our primary focus in this paper is on zero-shot inference, TC can be also applied to few-shot scenarios. In Figure 4, we report n-shot (n ranges from 1 to 4) results

Table 2: Results using Mistral-7b-Instruct-v0.3, Llama-2-7B-chat and Phi-3-mini-4k-instruct for zero-shot inference on 13 datasets. 'Original' indicates the LLM predictions without using any calibration method, which are determined by selecting the class with the highest probability. The best and second-best results are marked in bold fonts and ranked by color.

| Dataset | RTE | WNLI | SciTail | CB | MNLI | QNLI | Persp. | IBM. | EZ. | IAM | VAST | PAWS | QQP |
|---|---|---|---|---|---|---|---|---|---|---|---|---|---|
| **Mistral-7B-Instruct-v0.3** | | | | | | | | | | | | | |
| Original | 74.4 | 70.4 | 60.5 | 60.7 | 66.4 | 74.8 | 58.0 | 58.0 | 31.1 | 78.0 | 44.3 | 58.4 | 50.6 |
| CC | 76.2 | **71.8** | 62.6 | 66.1 | **66.9** | 75.8 | 58.3 | 58.4 | 33.8 | 77.2 | 48.3 | **61.6** | 46.8 |
| DCPMI | **76.5** | 69.0 | **63.0** | 62.5 | 66.7 | **76.3** | 51.3 | 54.1 | 32.7 | 76.7 | 43.8 | 51.7 | **52.0** |
| DC | 73.6 | 70.4 | 58.4 | **73.2** | 64.7 | 72.4 | **64.0** | **60.1** | 33.8 | 77.2 | 47.7 | 58.4 | 49.7 |
| BC | 74.7 | 70.4 | 61.7 | 64.3 | 66.7 | 75.3 | 61.9 | 58.9 | **34.4** | **78.2** | **50.1** | 61.3 | 50.4 |
| **TC** | **78.0** | **73.2** | **64.3** | **82.1** | **68.1** | **77.8** | **65.4** | **69.8** | **36.0** | **79.5** | **49.4** | **63.0** | **54.9** |
| **Llama-2-7B-chat** | | | | | | | | | | | | | |
| Original | 53.1 | 43.7 | 39.9 | 46.4 | 37.6 | 49.5 | 42.8 | 43.7 | 22.1 | 51.4 | 22.3 | 44.2 | 53.2 |
| CC | 56.0 | 45.1 | 40.7 | 37.5 | 43.0 | 50.1 | 45.7 | 47.1 | 27.3 | 56.4 | **30.8** | 44.3 | 53.7 |
| DCPMI | 56.3 | 45.1 | 40.7 | 19.6 | 38.0 | 50.1 | 46.5 | 48.0 | 26.0 | 57.5 | 25.5 | **52.8** | 25.8 |
| DC | 56.0 | 57.7 | 48.6 | 42.9 | **46.8** | 56.6 | 49.9 | 48.4 | 21.0 | **65.5** | 22.1 | 44.4 | **54.0** |
| BC | **60.6** | **64.8** | 50.9 | 50.0 | 46.5 | 59.1 | 51.6 | 49.3 | 29.9 | 60.3 | 30.3 | 52.2 | 53.8 |
| **TC** | 57.0 | 62.0 | **63.4** | **55.4** | 45.3 | **64.8** | 52.0 | **52.3** | 30.4 | 57.5 | **31.1** | 58.5 | 55.3 |
| **Phi-3-mini-4k-instruct** | | | | | | | | | | | | | |
| Original | 70.8 | 71.8 | 61.9 | 39.3 | 58.9 | 72.7 | 60.3 | 52.1 | 24.7 | 71.5 | 32.7 | 79.9 | 48.7 |
| CC | 69.7 | 71.8 | 62.7 | 10.7 | 36.6 | 71.4 | 51.0 | 45.4 | 28.6 | 71.0 | 40.3 | 78.8 | 45.8 |
| DCPMI | 71.1 | **76.1** | 55.3 | **76.8** | 54.5 | **75.0** | 41.3 | 39.2 | **37.8** | **73.4** | 47.7 | 80.9 | 50.0 |
| DC | 72.2 | 66.2 | 49.2 | 64.3 | **66.8** | 66.2 | 59.9 | 55.4 | 36.7 | 71.3 | 39.5 | **81.8** | 51.8 |
| BC | 71.1 | 73.2 | **65.9** | 64.3 | **63.7** | 74.8 | **64.4** | **58.9** | 36.9 | 72.7 | **49.9** | **81.8** | 49.8 |
| **TC** | **73.6** | **74.6** | **64.3** | **83.9** | 59.9 | **78.5** | **66.9** | **66.0** | **39.4** | 75.7 | **51.9** | **83.0** | **54.7** |

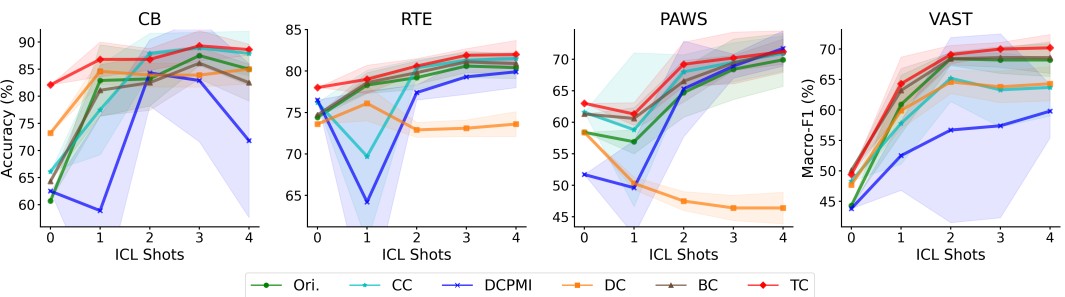

Figure 4: The few-shot performance of Mistral-7B-Instruct-v0.3 using various calibration methods over the number of in-context learning (ICL) shots. Lines and shades denote the mean and standard deviation, respectively, for 5 randomly sampled sets used for few-shot inference.

of Mistral-7b-Instruct-v0.3 on CB, RTE, PAWS and VAST datasets. We present the average results of five randomly sampled sets of n examples drawn from the training set, along with their standard deviations. The overall trend reveals that our proposed TC again outperforms baseline methods on these datasets with low variance, indicating its strong generalization ability. We also observe a general trend of improved performance with an increased number of shots, and the performance gap between TC and original LLM suggests that TC enables LLMs to more effectively leverage in-context demonstrations.

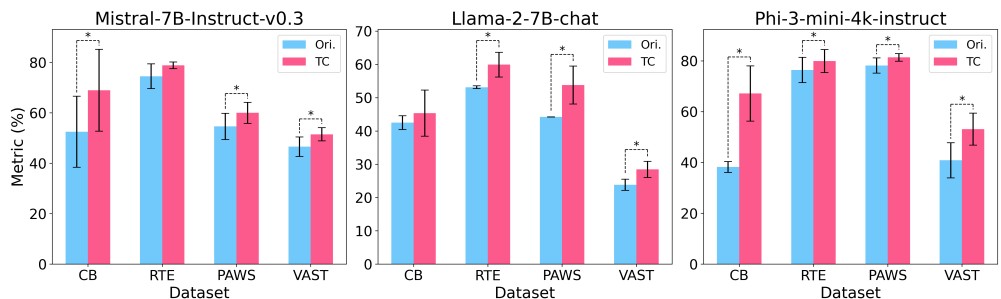

Figure 5: The means and standard deviations over the five different templates considered for CB, RTE, PAWS and VAST datasets. '*' indicates the significant improvement in performance over the original LLM (paired t-test with p ≤ 0.05).

Table 3: Zero-shot performance of Mistral-7b-Instruct-v0.3 and Phi-3-mini-4k-instruct on additional sentiment analysis, offensive language identification and hate speech detection tasks. The best and second-best results are marked in bold fonts and ranked by color.

| Model | Mistral-7B-Instruct-v0.3 | | | | | | Phi-3-mini-4k-instruct | | | | | |
|---|---|---|---|---|---|---|---|---|---|---|---|---|
| Method | Ori. | CC | DCPMI | DC | BC | TC | Ori. | CC | DCPMI | DC | BC | TC |
| SST-2 | 83.9 | 81.7 | 80.7 | **85.0** | 84.3 | **86.8** | 77.4 | 74.0 | 85.8 | **89.8** | 82.7 | 89.0 |
| OffensEval | 58.3 | 55.2 | 53.2 | **59.4** | 58.3 | **61.7** | 43.6 | 42.3 | 46.4 | **56.3** | 56.3 | 63.5 |
| HatEval | 61.2 | 60.1 | 59.6 | **62.3** | 62.2 | **66.5** | 36.7 | 36.6 | 37.0 | 54.6 | **55.9** | 63.5 |
| HateSpeech18 | 55.2 | 54.6 | 54.3 | **57.7** | 56.2 | **70.9** | 33.8 | 33.8 | 34.3 | 41.9 | **44.3** | 61.0 |

## 6.2 EFFECTIVENESS ANALYSIS

We conduct more experiments to verify the effectiveness of TC. The evaluation is performed under the zero-shot setting for all experiments.

**Robustness.** We conduct the experiments across five different prompt templates (details of templates are shown in Table 8 of Appendix), and report the means and standard deviations on CB, RTE, PAWS and VAST datasets. In Figure 5, we observe that TC shows consistent improvements over the original LLM, often by a hefty margin, indicating that TC is more effective and robust to various prompt templates. In addition, the results show that the model exhibits better performance with specific templates, which suggests that a well-designed prompt template can further improve the performance of TC. Overall, TC strengthens the stability of LLM predictions with regard to prompt designs, thereby simplifying the task of prompt engineering.

**Other NLU Tasks.** To assess the generalization ability of TC, besides the inference tasks mentioned in Table 2, we consider three additional NLU tasks (sentiment analysis, offensive language identification and hate speech detection) for evaluation. We reformulate the task definition to align with the format of NLI. For example, with the HateSpeech18 dataset, we utilize the original input text as the premise and take "the text expresses hate speech." as the hypothesis. The details of prompt templates are shown in Table 7 of Appendix. Table 3 shows the performance of Mistral-7B-Instruct-v0.3 and Phi-3-mini-4k-instruct on these tasks. We observe that TC improves the original LLM by an average of 6.8% and 21.4% on Mistral-7B-Instruct-v0.3 and Phi-3-mini-4k-instruct models, respectively. Furthermore, TC shows remarkable improvements over calibration methods on these datasets. It suggests that TC significantly mitigates the inherent bias of LLMs, highlighting its potential as a universally applicable method for addressing such bias across diverse tasks. We also compare TC with baselines that directly prompt LLMs for classification, and results are shown in Table 9 of Appendix.

## 6.3 BIAS ANALYSIS

Though previous calibration methods have demonstrated better performance over the original LLM, we argue that these methods are not always optimal, which may not effectively mitigate the prefer-

Table 4: Experimental results of zero-shot inference with TC using Mistral-7B-Instruct-v0.3, Llama-2-7B-chat and Phi-3-mini-4k-instruct models. '+TC' indicates the combination of TC with the previous calibration method. The best results are marked in bold fonts. Underlined scores indicate that baseline+TC shows improvements over TC.

| Dataset | RTE | WNLI | SciTail | CB | MNLI | QNLI | Persp. | IBM. | EZ. | IAM | VAST | PAWS | QQP |
|---|---|---|---|---|---|---|---|---|---|---|---|---|---|
| **Mistral-7B-Instruct-v0.3** | | | | | | | | | | | | | |
| TC | 78.0 | 73.2 | 64.3 | 82.1 | 68.1 | 77.8 | 65.4 | 69.8 | 36.0 | 79.5 | 49.4 | 63.0 | 54.9 |
| CC | 76.2 | 71.8 | 62.6 | 66.1 | 66.9 | 75.8 | 58.3 | 58.4 | 33.8 | 77.2 | 48.3 | 61.6 | 46.8 |
| +TC | 78.3 | 74.6 | 64.5 | 82.1 | 68.0 | 78.2 | 65.5 | 69.9 | 36.3 | 79.3 | 50.0 | 63.5 | 55.0 |
| DCPMI | 76.5 | 69.0 | 63.0 | 62.5 | 66.7 | 76.3 | 51.3 | 54.1 | 32.7 | 76.7 | 43.8 | 51.7 | 52.0 |
| +TC | 78.3 | 74.6 | 64.7 | 80.4 | 67.8 | 78.5 | 64.0 | 69.4 | 34.0 | 79.3 | 48.5 | 62.2 | 54.8 |
| DC | 73.6 | 70.4 | 58.4 | 73.2 | 64.7 | 72.4 | 64.0 | 60.1 | 33.8 | 77.2 | 47.7 | 58.4 | 49.7 |
| +TC | 78.0 | 74.6 | 56.3 | 83.9 | 65.4 | 78.7 | 66.4 | 70.2 | 35.9 | 79.5 | 48.3 | 63.2 | 55.0 |
| BC | 74.7 | 70.4 | 61.7 | 64.3 | 66.7 | 75.3 | 61.9 | 58.9 | 34.4 | 78.2 | 50.1 | 61.3 | 50.4 |
| +TC | 77.6 | 74.6 | 65.4 | 69.6 | 68.8 | 78.0 | 66.6 | 68.0 | 38.5 | 78.6 | 50.3 | 63.7 | 55.0 |
| **Llama-2-7B-chat** | | | | | | | | | | | | | |
| TC | 57.0 | 62.0 | 63.4 | 55.4 | 45.3 | 64.8 | 52.0 | 52.3 | 30.4 | 57.5 | 31.1 | 58.5 | 55.3 |
| CC | 56.0 | 45.1 | 40.7 | 37.5 | 43.0 | 50.1 | 45.7 | 47.1 | 27.3 | 56.4 | 30.8 | 44.3 | 53.7 |
| +TC | 56.3 | 63.4 | 63.6 | 55.4 | 47.4 | 64.7 | 52.3 | 52.8 | 31.5 | 57.3 | 31.9 | 58.5 | 55.2 |
| DCPMI | 56.3 | 45.1 | 40.7 | 19.6 | 38.0 | 50.1 | 46.5 | 48.0 | 26.0 | 57.5 | 25.5 | 52.8 | 25.8 |
| +TC | 56.7 | 63.4 | 63.6 | 46.4 | 47.0 | 64.8 | 52.4 | 53.0 | 30.4 | 57.3 | 30.3 | 58.9 | 54.7 |
| DC | 56.0 | 57.7 | 48.6 | 42.9 | 46.8 | 56.6 | 49.9 | 48.4 | 21.0 | 65.5 | 22.1 | 44.4 | 54.0 |
| +TC | 59.9 | 60.6 | 57.2 | 44.6 | 46.8 | 65.7 | 52.6 | 52.6 | 24.3 | 60.5 | 25.0 | 51.3 | 55.5 |
| BC | 60.6 | 64.8 | 50.9 | 50.0 | 46.5 | 59.1 | 51.6 | 49.3 | 29.9 | 60.3 | 30.3 | 52.2 | 53.8 |
| +TC | 66.1 | 66.2 | 57.7 | 53.6 | 47.7 | 67.5 | 53.1 | 53.6 | 33.6 | 64.5 | 30.8 | 58.3 | 55.7 |
| **Phi-3-mini-4k-instruct** | | | | | | | | | | | | | |
| TC | 73.6 | 74.6 | 64.3 | 83.9 | 59.9 | 78.5 | 66.9 | 66.0 | 39.4 | 75.7 | 51.9 | 83.0 | 54.7 |
| CC | 69.7 | 71.8 | 62.7 | 10.7 | 36.6 | 71.4 | 51.0 | 45.4 | 28.6 | 71.0 | 40.3 | 78.8 | 45.8 |
| +TC | 72.9 | 74.6 | 64.7 | 83.9 | 58.8 | 78.6 | 66.7 | 66.0 | 39.2 | 75.7 | 52.6 | 83.0 | 54.7 |
| DCPMI | 71.1 | 76.1 | 55.3 | 76.8 | 54.5 | 75.0 | 41.3 | 39.2 | 37.8 | 73.4 | 47.7 | 80.9 | 50.0 |
| +TC | 74.0 | 73.2 | 63.0 | 83.9 | 59.0 | 78.0 | 66.1 | 66.1 | 37.5 | 75.3 | 44.4 | 83.0 | 54.7 |
| DC | 72.2 | 66.2 | 49.2 | 64.3 | 66.8 | 66.2 | 59.9 | 55.4 | 36.7 | 71.3 | 39.5 | 81.8 | 51.8 |
| +TC | 73.6 | 69.0 | 61.3 | 78.6 | 67.8 | 79.9 | 66.9 | 67.8 | 34.9 | 75.5 | 37.8 | 82.9 | 55.1 |
| BC | 71.1 | 73.2 | 65.9 | 64.3 | 63.7 | 74.8 | 64.4 | 58.9 | 36.9 | 72.7 | 49.9 | 81.8 | 49.8 |
| +TC | 72.6 | 76.1 | 65.4 | 78.6 | 69.2 | 81.8 | 68.2 | 68.4 | 39.0 | 74.8 | 52.4 | 82.5 | 54.1 |

ence bias in inference tasks. To further substantiate our claim, we conduct additional experiments by applying each previous calibration method to predictions used in TC. For example, we first calibrate the $p(y|x_p)$, $p(y|x_h)$ and $p(y|x_p, x_h)$ with BC, and then perform the task calibration. Experimental results of three LLMs are shown in Table 4. We find that almost all baseline methods exhibit improved performance with TC on three models, as evidenced by the bold numbers in the table. Compared to CC, DCPMI, and DC relying on content-free tokens that may introduce additional biases (Zhou et al., 2024), TC encourages the model to reason based on both premise and hypothesis, thereby achieving superior bias mitigation. BC computes the correction term once after all test samples are seen, whereas TC computes the $p(y|x_p)$ and $p(y|x_h)$ for each sample, which can be seen as a more general instance-specific approach for calibration. In addition, we can also observe that baseline+TC outperforms TC on multiple datasets, which indicates that contributions from task reformulation do not fully overlap with previous methods on reducing the bias. We leave the further exploration of integrating TC with other calibration methods in future work.

Table 5: Examples of applying task calibration to predictions of Phi-3-mini-4k-instruct. 'Ori.' indicates the original LLM prediction using both the sentence and the question as input. 'S' and 'Q' indicate LLM predictions using only the sentence and the question, respectively. All samples are taken from QNLI dataset (Rajpurkar et al., 2016). Correct answers are highlighted in bold.

| | Sentence | Question | Ori. | S | Q | TC |
|---|---|---|---|---|---|---|
| 1 | In Afghanistan, the mujahideen's victory against the Soviet Union in the 1980s did not lead to justice and prosperity, due to a vicious and destructive civil war between political and tribal warlords, making Afghanistan one of the poorest countries on earth. | What did the civil war leave the state of Afghanistan's economy in? | false | **true** | false | **true** |
| 2 | Unlike a traditional community pharmacy where prescriptions for any common medication can be brought in and filled, specialty pharmacies carry novel medications that need to be properly stored, administered, carefully monitored, and clinically managed. | Besides drugs, what else do specialty pharmacies provide? | true | true | true | **false** |
| 3 | Although parts of Sunnyside are within the City of Fresno, much of the neighborhood is a "county island" within Fresno County. | Where is the neighborhood of Sunnyside located in Fresno? | true | **false** | **false** | true |

## 6.4 CASE STUDIES

To get a better impression of how TC works, we perform an in-depth analysis on QNLI and present three examples in Table 5. Correct answers are highlighted in bold. Results show that TC accurately predicts 61% of the instances that were initially misclassified by the original LLM using both the sentence and the question as input on QNLI (Ex. 1-2). In the second example, despite the incorrect predictions of 'Original', 'S' and 'Q', TC successfully identifies the correct label *false*, which demonstrates the effectiveness of reducing LLMs' reliance on individual component (i.e., the sentence or the question) at inference time. However, we also observe that TC encounters failure in some rare cases (Ex. 3), accounting for approximately 5% of the erroneous predictions by the original LLM. As shown in the third example, TC fails to correct the LLM prediction when both 'S' and 'Q' provide the accurate predictions. Overall, we see that TC can effectively calibrate LLM predictions by utilizing the predictions of the premise (sentence) and the hypothesis (question).

## 7 CONCLUSION AND LIMITATIONS

We proposed task calibration (TC), a zero-shot and inference-only calibration method that reformulates inference tasks to mitigate the effects of spurious correlations. Experimental results show that TC achieves state-of-the-art performance on 13 inference datasets under zero-shot setting. Furthermore, our method demonstrates its effectiveness in few-shot settings and other NLU tasks such as hate speech detection. TC is also robust to various prompt templates and has the potential to be integrated with other calibration methods. To our knowledge, we are the first to consider the synergistic effect of premise and hypothesis over their individual effects in model calibration.

A limitation of our proposed method is that it requires extra computational cost owing to the use of premise-only and hypothesis-only predictions at inference time, which could be alleviated with model acceleration techniques such as pruning and quantization. In addition, our method may not be fully compatible with closed-source LLMs such as GPT-4 and Claude-3 due to the potential lack of access to prediction logits, which is also prevalent among most previous calibration methods. We acknowledge that this is not an exhaustive study on all existing tasks, where further exploration of extending our method to more diverse NLP tasks should be done in future work.

## REPRODUCIBILITY STATEMENT

To ensure the reproducibility of our results, we have made detailed efforts throughout the paper. All experimental setups, including benchmarks, the implementation of previous baselines, and model details, are described in Section 3. In addition, we provide detailed dataset statistics in Appendix A and present all prompt templates in Appendix B. Our code and data will be made publicly available upon publication.

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

## A    DATASET STATISTICS

In the main experiments, we use 13 datasets falling into three categories: natural language inference, stance detection and paraphrasing. We additionally consider sentiment analysis, offensive language identification and hate speech detection to indicate the effectiveness of TC. We use the test set for evaluation except for GLUE (Wang et al., 2018) and SuperGLUE (Wang et al., 2019) datasets (i.e., RTE, WNLI, CB, MNLI, QNLI, QQP and SST-2), for which we use the full validation set for evaluation. We summarize the dataset statistics in Table 6.

## B    PROMPT TEMPLATES

We show the templates and label names for all datasets in Table 7. For NLI tasks, we follow the previous works (Holtzman et al., 2021; Fei et al., 2023) and use *true/false/neither* as the label set. For stance detection tasks, we use *favor/against/neutral* as the label set, which is consistent with previous works (Zhang et al., 2022; Zhao et al., 2024). The label *neither* or *neutral* is removed from the label set for the binary classification tasks.

In addition, we show the templates and label names used in robustness experiments in Table 8. Besides the original prompt as shown in Table 7, we introduce four additional templates and label sets for each dataset to verify the robustness of TC towards various templates on inference tasks.

## C    DIRECT PROMPTING FOR CLASSIFICATION TASKS

Besides the experimental setting of task reformulation as discussed in Section 6.2, we also compare TC with baselines in the setting of direct prompting. We follow the prompt templates and label sets of previous work (Fei et al., 2023; Zhou et al., 2024). Table 9 shows the performance of Mistral-7B-Instruct-v0.3 and Phi-3-mini-4k-instruct under this setting. Results indicate that TC still achieves the best performance on all datasets, which further validate our claim that TC has the potential to be a universally applicable method for addressing spurious correlations across diverse tasks.

## D    AN ENSEMBLE OF PREMISE AND HYPOTHESIS CALIBRATION

We also consider ensembling the results of premise calibration and hypothesis calibration using batch calibration (BC). Specifically, we individually calibrate premise and hypothesis predictions

Table 6: Details of the dataset used for evaluation in the Table 2. #Test denotes the number of test samples. We consistently use the validation split as the test split for datasets where test labels are not publicly available.

| Dataset | Task | #Class | #Test |
|---|---|---|---|
| RTE | Natural Language Inference | 2 | 277 |
| WNLI | Natural Language Inference | 2 | 71 |
| SciTail | Natural Language Inference | 2 | 2,126 |
| CB | Natural Language Inference | 3 | 56 |
| MNLI-M | Natural Language Inference | 3 | 9,815 |
| MNLI-MM | Natural Language Inference | 3 | 9,832 |
| QNLI | Natural Language Inference | 2 | 5,463 |
| Perspectrum | Stance Detection | 2 | 2,773 |
| IBM30K | Stance Detection | 2 | 6,315 |
| EZ-Stance | Stance Detection | 3 | 7,798 |
| IAM | Stance Detection | 2 | 527 |
| VAST | Stance Detection | 3 | 1,460 |
| PAWS | Paraphrasing | 2 | 8,000 |
| QQP | Paraphrasing | 2 | 40,430 |
| SST-2 | Sentiment Analysis | 2 | 872 |
| OffensEval | Offensive Language Identification | 2 | 860 |
| HatEval | Hate Speech Detection | 2 | 2,970 |
| HateSpeech18 | Hate Speech Detection | 2 | 478 |

using BC and then aggregate the outputs. Results are shown in Table 10. We can observe that TC significantly outperforms this baseline (which we call BC-en) on all datasets across three LLMs, which indicates the importance of the proposed mutual information method. The performance of BC-en is worse than BC because NLI tasks require both premise and hypothesis information to infer the entailment label.

Table 7: Prompt templates for the main experiments on each task. The inputs are marked in {}.

| Dataset | Template | Label |
|---|---|---|
| RTE | {Premise} entails {Hypothesis}. true or false? Answer: {Label} | true/false |
| WNLI | {Text 1} entails {Text 2}. true or false? Answer: {Label} | true/false |
| SciTail | {Premise} entails {Hypothesis}. true or false? Answer: {Label} | true/false |
| CB | {Premise}. Hypothesis: {Hypothesis}. true, false or neither? Answer: {Label} | true/false/neither |
| MNLI | {Premise}. Hypothesis: {Hypothesis}. true, false or neither? Answer: {Label} | true/false/neither |
| QNLI | {Text} contains the answer to {Question}. true or false? Answer: {Label} | true/false |
| Perspectrum | What is the stance of {Text} on {Target}? favor, against or neutral? Answer: {Label} | favor/against/neutral |
| IBM30K | What is the stance of {Text} on {Target}? favor, against or neutral? Answer: {Label} | favor/against/neutral |
| EZ-Stance | What is the stance of {Text} on {Target}? favor, against or neutral? Answer: {Label} | favor/against/neutral |
| IAM | {Claim} gives a favorable answer to {Topic}? true or false? Answer: {Label} | true/false |
| VAST | What is the stance of {Text} on {Target}? favor, against or neutral? Answer: {Label} | favor/against/neutral |
| PAWS | Sentence 1: {Text 1}. Sentence 2: {Text 2}. Duplicate: true or false? Answer: {Label} | true/false |
| QQP | Question 1: {Text 1}. Question 2: {Text 2}. Duplicate: true or false? Answer: {Label} | true/false |
| SST-2 | {Text} entails {Claim}. true or false? Answer: {Label} | true/false |
| OffensEval | {Text} entails {Claim}. true or false? Answer: {Label} | true/false |
| HatEval | {Text} entails {Claim}. true or false? Answer: {Label} | true/false |
| HateSpeech18 | {Text} entails {Claim}. true or false? Answer: {Label} | true/false |

Table 8: Prompt templates for the robustness experiments on RTE, CB, VAST and PAWS datasets. The inputs are marked in {}.

| Dataset | ID | Template | Label |
|---|---|---|---|
| RTE | 1 | {Premise} entails {Hypothesis}. true or false? Answer: {Label} | true/false |
| | 2 | {Premise}. Hypothesis: {Hypothesis}. true or false? Answer: {Label} | true/false |
| | 3 | {Premise}. Question: {Hypothesis}. true or false? Answer: {Label} | true/false |
| | 4 | {Premise}. Question: {Hypothesis}. entailment or contradiction? Answer: {Label} | entailment/ contradiction |
| | 5 | Does the premise {Premise} entail the hypothesis {Hypothesis}? yes or no? Answer: {Label} | yes/no |
| CB | 1 | {Premise} entails {Hypothesis}. true, false or neither? Answer: {Label} | true/false/neither |
| | 2 | {Premise}. Hypothesis: {Hypothesis}. true, false or neither? Answer: {Label} | true/false/neither |
| | 3 | {Premise}. Question: {Hypothesis}. true, false or neither? Answer: {Label} | true/false/neither |
| | 4 | {Premise}. Question: {Hypothesis}. entailment, contradiction or neutral? Answer: {Label} | contradiction/ entailment/neutral |
| | 5 | Does the premise {Premise} entail the hypothesis {Hypothesis}? yes, no or neither? Answer: {Label} | yes/no/neither |
| VAST | 1 | What is the stance of {Text} on {Target}? favor, against or neutral? Answer: {Label} | favor/against/neutral |
| | 2 | What is the attitude of the sentence {Text} towards {Target}? favor, against or neutral? Answer: {Label} | favor/against/neutral |
| | 3 | Does {Text} support {Target}? true, false or neither? Answer: {Label} | true/false/neither |
| | 4 | {Text} supports {Target}. true, false or neither? Answer: {Label} | true/false/neither |
| | 5 | Sentence: {Text}. Target: {Target}. Stance: favor, against or neutral? Answer: {Label} | favor/against/neutral |
| PAWS | 1 | Sentence 1: {Text 1}. Sentence 2: {Text 2}. Duplicate: true or false? Answer: {Label} | true/false |
| | 2 | Sentence 1: {Text 1}. Sentence 2: {Text 2}. Is Sentence 2 the duplicate of Sentence 1? true or false? Answer: {Label} | true/false |
| | 3 | Text 1: {Text 1}. Text 2: {Text 2}. Duplicate: true or false? Answer: {Label} | true/false |
| | 4 | Sentence 1: {Text 1}. Sentence 2: {Text 2}. Equivalence: true or false? Answer: {Label} | true/false |
| | 5 | Sentence 1: {Text 1}. Sentence 2: {Text 2}. Duplicate: yes or no? Answer: {Label} | yes/no |

Table 9: Zero-shot performance of Mistral-7b-Instruct-v0.3 and Phi-3-mini-4k-instruct on additional sentiment analysis, offensive language identification and hate speech detection tasks in the direct prompting setting. The best and second-best results are marked in bold fonts and ranked by color.

| Model | Mistral-7B-Instruct-v0.3 | | | | | | Phi-3-mini-4k-instruct | | | | | |
|---|---|---|---|---|---|---|---|---|---|---|---|---|
| Method | Ori. | CC | DCPMI | DC | BC | TC | Ori. | CC | DCPMI | DC | BC | TC |
| SST-2 | 72.9 | 75.3 | 82.8 | 81.7 | **83.1** | **86.8** | 84.9 | 84.1 | 84.1 | 84.1 | 84.6 | **89.0** |
| OffensEval | 52.9 | 36.9 | 41.0 | **57.7** | 53.6 | **61.7** | 41.8 | **42.6** | 36.1 | 41.3 | 42.4 | **63.5** |
| HatEval | 48.3 | 34.8 | 38.4 | 60.2 | **61.7** | **66.5** | 49.2 | **49.9** | 46.0 | **49.9** | 49.9 | **63.5** |
| HateSpeech18 | 63.6 | 48.9 | 53.7 | 67.5 | **69.3** | **70.9** | 59.4 | 57.9 | 59.7 | **60.2** | 59.9 | **61.0** |

Table 10: Comparison of TC with BC-en using Mistral-7b-Instruct-v0.3, Llama-2-7B-chat and Phi-3-mini-4k-instruct for zero-shot inference on 13 datasets. The best results are marked in bold fonts.

| Dataset | RTE | WNLI | SciTail | CB | MNLI | QNLI | Persp. | IBM. | EZ. | IAM | VAST | PAWS | QQP |
|---|---|---|---|---|---|---|---|---|---|---|---|---|---|
| **Mistral-7B-Instruct-v0.3** | | | | | | | | | | | | | |
| BC | 74.7 | 70.4 | 61.7 | 64.3 | 66.7 | 75.3 | 61.9 | 58.9 | 34.4 | 78.2 | **50.1** | 61.3 | 50.4 |
| BC-en | 59.2 | 49.3 | 46.9 | 25.0 | 36.0 | 49.1 | 51.8 | 38.5 | 27.7 | 57.9 | 37.3 | 47.7 | 33.4 |
| TC | **78.0** | **73.2** | **64.3** | **82.1** | **68.1** | **77.8** | **65.4** | **69.8** | **36.0** | **79.5** | 49.4 | **63.0** | **54.9** |
| **Llama-2-7B-chat** | | | | | | | | | | | | | |
| BC | **60.6** | **64.8** | 50.9 | 50.0 | **46.5** | 59.1 | 51.6 | 49.3 | 29.9 | **60.3** | 30.3 | 52.2 | 53.8 |
| BC-en | 53.4 | 52.1 | 44.8 | 42.9 | 37.7 | 50.2 | 49.8 | 48.8 | 29.8 | 53.1 | 30.0 | 47.7 | 48.8 |
| TC | 57.0 | 62.0 | **63.4** | **55.4** | 45.3 | **64.8** | **52.0** | **52.3** | **30.4** | 57.5 | **31.1** | **58.5** | **55.3** |
| **Phi-3-mini-4k-instruct** | | | | | | | | | | | | | |
| BC | 71.1 | 73.2 | **65.9** | 64.3 | **63.7** | 74.8 | 64.4 | 58.9 | 36.9 | 72.7 | 49.9 | 81.8 | 49.8 |
| BC-en | 56.7 | 57.7 | 56.0 | 26.8 | 35.7 | 49.9 | 55.4 | 42.4 | 30.6 | 64.9 | 38.1 | 51.9 | 43.6 |
| TC | **73.6** | **74.6** | 64.3 | **83.9** | 59.9 | **78.5** | **66.9** | **66.0** | **39.4** | **75.7** | **51.9** | **83.0** | **54.7** |

