# OpenReview forum: "Task Calibration: Calibrating Large Language Models on Inference Tasks"
_ICLR.cc/2025/Conference — Submitted to ICLR 2025_

### Official Review · Reviewer_xLua · 2024-10-24

**Soundness:** 4
**Presentation:** 3
**Contribution:** 2
**Rating:** 5
**Confidence:** 5

**Summary:**

This paper proposes a new calibration method for natural language inference via generative language models. The authors first identify the premise side spurious correlation inside natural language inference and verify its existence inside generative natural language inference. Based on the validation of the issue, the authors propose to use mutual information between the premise and the hypothesis as a calibration factor to improve the accuracy of natural language inference, which shows improvement on multiple datasets and multiple models.

**Strengths:**

- The paper proposes a new calibration method for natural language inference via generative language models, which has been shown to be promising by experiments.

- The method is experimented on comprehensive datasets and models, which makes the conclusion solid.

**Weaknesses:**

- While the author is claiming the discovery of premise side spurious correlation to be an important contribution, many previous works have studied the hypothesis side spurious correlation (also as cited). There is not significant difference between the roles of premise and hypothesis in natural language inference, which makes the contribution of this discovery incremental.

- The studied paradigm is a bit too narrow, which improves a method of solving a specific task (natural language inference). Different from baselines, the method is only applicable when there are two input factors.

- The paper lacks baselines using premise calibration. Based on the discovery of premise side spurious correlation, the most straightforward way to address the issue should be ensembling the score from premise calibration and hypothesis calibration, which is not included in the comparison to show the importance of the proposed mutual information method. **(addressed)**

- At this point, the studied paradigm deviates a bit from the mainstream of how language models make inferences with chain-of-thoughts. The authors should discuss how the calibration for direct classification can be adapted to paradigms that generate chain-of-thoughts before making the classification. **(addressed)**

**Questions:**

My problems are listed in the weakness part, I also have the following questions for the authors,

- The performance of Llama-2-7B-chat seems a bit too weak, can you provide some explanations about this? **(addressed)**

- The performance of all models on QQP is also too weak, as QQP is a semantic similarity benchmark, are you using the correct prompt/verbalizer in the evaluation? **(addressed)**

- The performance in Table 3 is not compared with direct prompting the language model for classification, can you explain the absence of these baselines? **(addressed)**

---

### Official Review · Reviewer_jCfw · 2024-11-04

**Soundness:** 4
**Presentation:** 4
**Contribution:** 4
**Rating:** 10
**Confidence:** 3

**Summary:**

This paper introduced a method that uses mutual information to change the inference scoring function when generating tokens to calibrate LLMs for better inference, considering inputs and label correlation biases produced during LLM training.

**Strengths:**

1. The paper has enough novelty. Although mutual information is not new, applying it to the inference score function can be considered novel.
2. It includes all the previously related works and lists the differences.
3. The paper writing is clear, and the visuals are good.
4. It has detailed experiments and results analysis.

**Weaknesses:**

N/A

**Questions:**

N/A

---

### Official Review · Reviewer_GH1T · 2024-11-04

**Soundness:** 3
**Presentation:** 3
**Contribution:** 3
**Rating:** 6
**Confidence:** 5

**Summary:**

In this paper, the authors propose a calibration strategy for NLI based tasks. This calibration strategy runs in inference time, requiring no modification of the model or performance dip. The authors claim that this approach mitigates some structural biases that are exhibited by LLMs for NLP tasks. They also claim that this approach is not sensitive to prompt templates. The authors compare it to several existing calibration methods to show that their approach is better.

**Strengths:**

The approach is simple and if the results hold, might be a useful method to calibrate LLMs for NLI based reasoning tasks.

**Weaknesses:**

The paper has several flaws:

For motivation, the paper cites papers such as Gururangan et al (2018), which study biases in NLI models and papers such as McKenna et al (2023) that studies a different bias in LLMs for NLI tasks. While the former work is done in models fine-tuned for NLI, the latter shows evidence for specific biases in terms of memorization and term frequency. This is a misleading equivalence in the introduction section. This paper would have benefitted from analyzing the biases in McKenna et al (2023) which seems to be closest in experimental setting. The specific biases that the authors introduce in the introduction which were based on older studies need to be established in the latest LLMs before claiming that these biases still exist in a meaningful way.  **(addressed)**

The experimental setup of “premise” only or “hypothesis” only is a bit confusing especially for tasks that are not NLI based. Why is a dataset like SST-2 used as NLI ? and how is it a valid way to ascertain model performance on this task? I would like to understand the authors’ reasoning on this part. The prompt formulation also masks whether the reported results are valid performance numbers of the task for a given model **(addressed)**

The models tested in this paper are Instruction-tuned models. Is there a specific reason that this choice was made ? I would like to know the reasoning behind this as well. Why not pretrained checkpoints of the models ? **(addressed)**

**Questions:**

Covered above

---

### Meta-Review · Area_Chair_ZiZD · 2024-12-12

**Metareview:**

This paper introduces Task Calibration (TC), a method to enhance LLM reasoning by balancing reliance on the premise and hypothesis, addressing spurious correlations, and improving zero-shot and few-shot performance across various tasks. However, the approach's applicability is narrow, and the contribution of identifying premise-side spurious correlations is incremental. While some concerns have been addressed, I still recommend a further round of review before considering acceptance. Therefore, I recommend rejecting this submission.

**Additional Comments On Reviewer Discussion:**

The authors have added numerous experiments to address Reviewer GH1T's concerns, which may warrant another round of review for this paper.

However, the rating provided by Reviewer jCfw seems abnormal, as no actionable suggestions were offered.

While the authors partially addressed the concerns raised by Reviewer xLua, the limited applicability of the proposed method, as highlighted by Reviewer xLua, remains a significant issue. I concur with this assessment and have therefore decided to reject the paper.

---

### Decision · Program_Chairs · 2025-01-22

Reject